# Childhood and Adolescent Obesity with Somatic Indicators of Stress, Inflammation, and Dysmetabolism before and after Intervention: A Meta-Analysis

**DOI:** 10.3390/jpm13091322

**Published:** 2023-08-28

**Authors:** Konstantina Dragoumani, Andreas Troumbis, Flora Bacopoulou, George Chrousos

**Affiliations:** 1Department of Food Science and Nutrition, University of the Aegean, 81400 Lemnos, Greece; 2Department of Environmental Studies, University of the Aegean, 81100 Mytilene, Greece; 3Center for Adolescent Medicine and UNESCO Chair in Adolescent Health Care, First Department of Pediatrics, School of Medicine, Aghia Sophia Children’s Hospital, National and Kapodistrian University of Athens, 11527 Athens, Greece; 4University Research Institute of Maternal and Child Health & Precision Medicine, Aghia Sophia Children’s Hospital, National and Kapodistrian University of Athens, 11527 Athens, Greece

**Keywords:** children, adolescents, overweight, obesity, dysmetabolism, inflammation

## Abstract

There have been numerous attempts to establish a correlation between obesity and stress, inflammatory, and dysmetabolism biomarkers in children and adolescents. Here, we performed a meta-analysis of existing studies to shed light on the elusive correlations of childhood and adolescent obesity with physiological indicators of stress, inflammation, and metabolism before and after lifestyle interventions. Observational studies, meta-analyses, narrative and systematic reviews were excluded. From a total of 53 articles, 11 were selected according to specific criteria. The biomarkers examined were circulating glucose, insulin, HDL, LDL, triglycerides, adiponectin, leptin, CRP, TNF-alpha, interleukin (IL)-6, systolic and diastolic blood pressure, and HOMA-IR. All analyses were performed using IBM SPSS Statistics Version 28.0.1.0 (142). The current meta-analysis provides evidence of a beneficial effect of a lifestyle intervention and/or drugs in children and adolescents living with obesity or overweight, consistent with a significant reduction in body fat—but not in BMI or waist circumference—an increase in circulating adiponectin and/or a reduction in serum insulin levels and diastolic blood pressure, and a trend towards a reduction of circulating leptin and glucose levels, as well as of the HOMA-IR. This meta-analysis indicates that lifestyle interventions could reduce overweight-/obesity-associated systemic inflammation and dysmetabolism even without an apparent decrease in BMI.

## 1. Introduction

Obesity is a complex medical condition that can lead to a wide range of health complications, many of which are linked to chronic low-grade systemic inflammation [1]. Chronic inflammation occurs when the body’s immune response is constantly activated at a low level over an extended period. Some complications associated with obesity and the potential issues that can result from chronic low-grade systemic inflammation are type 2 diabetes, cardiovascular disease, metabolic syndrome, non-alcoholic fatty liver disease (NAFLD), respiratory issues, joint problems, cancer, neurological effects, hormonal dysregulation, psychological effects, immune dysfunction, and accelerated aging [2,3,4,5,6,7,8,9,10,11,12,13]. Obesity affects children from early childhood to late adolescence and is very likely to continue in their adult life. Over 340 million children and adolescents aged 5–19 years were overweight or dealing with obesity in 2016. The prevalence of overweight and obesity among children and adolescents aged 5–19 years has risen dramatically from just 4% in 1975 to over 18% in 2016. The rise occurred similarly among both boys and girls: in 2016, 18% of girls and 19% of boys were overweight [14]. Obesity, amongst other definitions, could be described as abnormal or excessive fat accumulation that presents a major health hazard. It can be easily quantified via height and body weight percentiles in each age and gender group [15,16,17]. All in all, obesity is one of the most prevalent non-communicable diseases and a major concern for public health worldwide, mainly due to its well-established pathophysiologic manifestations, such as insulin resistance, diabetes mellitus type 2, atherosclerosis, hypertension, or even some types of cancer. Obesity has a strong, persistent underlying inflammatory component, represented by a state of low-grade chronic systemic inflammation that is known to be harmful to the metabolism, endocrinology, and overall homeostasis of human systems. Earlier observations of elevated circulating levels of inflammatory cytokines and other mediators of inflammation in individuals living with obesity imply that inflammation is the consequence of tilting the stress axis. Obesity is directly linked to the adipose tissue metabolomic and transcriptomic microenvironment as a multimodal vector composed of a repertoire of genetic, cellular, and organ-level information streams due to the accumulation of excessive fat and the inevitable cytokine secretion [18].

Lifestyle interventions play a crucial role in managing obesity and inflammation in children and adolescents. The most common changes in lifestyle consist in following a healthy diet and being active physically. Lifestyle interventions that promote a balanced, nutrient-rich diet can help reduce obesity and inflammation. Emphasizing the consumption of whole foods, fruits, vegetables, lean proteins, and healthy fats while reducing that of processed foods and sugary beverages can lead to weight loss and improved inflammatory markers [19,20]. Moreover, regular physical activity is associated with reduced obesity and improved inflammation in children and adolescents. Lifestyle interventions that encourage regular exercise can lead to weight management and the reduction of pro-inflammatory markers [21,22].

There have been numerous attempts in several studies to establish a correlation between obesity and stress indicators, inflammatory markers, or dysmetabolism markers in children and adults. Herein, an effort was made in the form of a meta-analysis to shed light on the elusive correlation of childhood and adolescent obesity with the above-mentioned physiological or molecular indicators and on how they can be modified by interventions focused on diet, exercise, supplements, or even drug therapy.

In this direction, a comprehensive review of the available scientific literature yielded over-abundant information, given that obesity-related literature is growing exponentially and numbers almost half a million articles to date. Special attention was therefore given to the filtering and careful selection of the available sources to allow for frequentist inference in childhood and adolescent obesity health effects. Although massively accumulated data might help search for statistical regularities in science, uncertainties, noise, and heterogenous (i.e., non-standardized) information might bias interpretation. Consequently, there is a need for an optimized mining pipeline to be deployed to search literature databases, as fragmented information is difficult to retrieve and combine towards a set hypothesis. In this study, we used a variety of in silico tools for creating bibliogram association networks [23] to navigate as safely as possible the available literature, using a combination of keywords and buzzwords that are essential to optimally access these large databases. We opted for establishing a keyword-oriented network revolving mainly around mesh terms (the NLM gold standard for indexing scientific articles). Keywords can be extracted from either the title or the abstract of any indexed publication, or they can be taken from the author-supplied list of keywords accompanying and following a certain publication. Bibliogram networks are used exactly because the earlier literature search mode that tended to focus on individual keywords is currently developing to include lexical units (i.e., sentences indicating a specific object).

In this study, a state-of-the-art combination of bibliometric software and tools and meta-analysis techniques was adopted, adapted, and used to collate published evidence and examine any potential correlation between children and/or adolescent obesity and specific markers/indicators before and upon intervention. The methodological hierarchy was positioned within the two views of medical science, with emphasis on the evaluation of existing data, especially from randomized trials, to “see whether there is “something” in it” [24] (p. 0339). After decades of meta-analyzing meta-analyses as a statistical technique in medical research, its pros and cons are well understood [25,26,27]. Several limitations have been already uncovered (e.g., replicability, sample size, statistical power effects, Type I and II errors, prior probabilities, and so on) and, potentially, are technically rectifiable [28,29,30]. We then explored the potential of coupling networks of medical objects–concepts with error-free meta-analysis techniques to construct a landscape of empirical knowledge regarding the childhood obesity epidemy.

A systematic meta-analysis investigating the relationship between stress, inflammation, and childhood obesity/overweight holds substantial scientific significance for various reasons. Firstly, it can provide a comprehensive overview of the existing research, clarifying the extent and consistency of the relationship between stress, inflammation, and childhood obesity/overweight. By pooling data from numerous studies, a meta-analysis offers a more precise estimation of the overall effect size and the strength of the relationships between stress, inflammation, and childhood obesity with respect to other types of analysis. Individual studies may produce conflicting results due to variations in methodology, populations, and other factors. A meta-analysis can help reconcile these discrepancies and offer a more balanced assessment of the relationship. Also, a meta-analysis can identify potential moderators that influence the stress–inflammation–obesity relationship, such as age, gender, socioeconomic status, and geographic location. This can lead to a more nuanced understanding of the phenomenon. A well-designed meta-analysis might provide insights into the temporal relationship between stress, inflammation, and obesity/overweight, aiding in the discussion about potential causal pathways. If a strong relationship is found, it can inform the clinical practice by highlighting the importance of addressing stress and inflammation in childhood obesity prevention and management strategies. It can identify gaps in the literature, areas of inconsistency, and areas with limited research. This information can guide future research efforts by highlighting the areas needing additional studies. If the relationship between stress, inflammation, and childhood obesity/overweight is robust, the findings can have implications for public health policies and interventions targeting childhood obesity and its underlying mechanisms. Finally, this information can advance the scientific knowledge by offering a synthesized perspective on a complex and evolving field, potentially leading to new hypotheses and research directions [31,32,33,34,35].

In summary, a systematic meta-analysis on the relationship between stress, inflammation, and childhood/adolescence obesity/overweight has the potential to provide comprehensive insights, refine our understanding, and guide future research. It offers a valuable synthesis of evidence that can inform clinical practice, interventions, and policy decisions related to childhood obesity and its underlying factors.

## 2. Materials and Methods

### 2.1. Literature Search Strategy

The following databases were searched from inception to November 2022: Web of Science, Scopus, and PubMed. Initially, no language or time frame restriction was applied. Several search strings, i.e., keyword and lexical unit combinations, were submitted to literature search engines, and the returns were noted for each database (see Appendix A). Two of them were selected from the several search strings to be further analyzed as bibliogram networks using the functionalities of the bibliometry VOSviewer software version 1.6.19. The detailed steps of the methodology used to visualize the conceptual landscape and identify its component variables are presented as a flow chart in Figure 1. The dependent, independent, and co-variant variables selected are reported in Table 1.

VOSviewer is a software tool for creating maps based on network data and visualizing and exploring these maps. The functionality of VOSviewer can be summarized as follows:Creating maps based on network data. A map can be created based on a network that is already available, but it is also possible to first construct a network. VOSviewer is an enhanced bibliometry visualization tool: it can be used to construct classical networks of scientific publications, journals, researchers, research organizations, countries, keywords, or terms. The items in these networks can be connected by co-authorship, co-occurrence, citation, bibliographic coupling, or co-citation links. To construct a bibliogrammatic network, bibliographic database files (i.e., Web of Science, Scopus, Dimensions, Lens, and PubMed files) and reference manager files (i.e., RIS, EndNote, and RefWorks files) can be provided as input to VOSviewer. Alternatively, VOSviewer can download data through an API (i.e., Crossref API, OpenAlex API, Europe PMC API, and several others). Most importantly, here, the VOSviewer was used as a platform to construct networks of lexical co-occurrences in texts as a data source.Visualizing and exploring maps. VOSviewer provides three types of map visualization: network visualization, overlay visualization, and density visualization. The zooming and scrolling functionality allows a map to be explored in full detail, which is essential when working with large maps containing thousands of items [36].

For the current study, bibliogrammatic networks were constructed as a graph of lexical unit co-occurrences based on title and abstract text data. The ultimate goals of this step were (1) to secure the accuracy of the search method; (2) to spot words or lexical units of main importance, and (3) to uncover new lexical conglomerates, if any, that might indicate different or alternative search strings, other than those selected initially.

Lastly, it is of uttermost importance to declare the aforementioned methodology was carefully designed and applied in full alignment with the PRISMA (Preferred Reporting Items for Systematic Reviews and Meta-Analyses) guidelines [37]. We formulated clear research objectives, established strict inclusion and exclusion criteria, and conducted a thorough screening of both titles and abstracts, as well as of full-text articles, in adherence to the PRISMA flow diagram. Data extraction was meticulously performed using standardized forms, while study quality and risk of bias were rigorously assessed according to the PRISMA recommendations. By employing appropriate statistical methods, we synthesized and analyzed the relevant data, assessed heterogeneity using the I^2^ statistic, and conducted sensitivity and subgroup analyses where appropriate, in alignment with the PRISMA guidelines. The potential for publication bias was evaluated through the methods outlined by PRISMA, and the implications of the findings were interpreted in the context of the research question. Our adherence to the PRISMA guidelines ensured transparency, reliability, and the highest standard of methodological rigor throughout the entire review process.

### 2.2. Selection Criteria

Observational studies, meta-analyses, narrative and systematic reviews, case reports, and case series were excluded. From the total 53 articles (after the removal of duplicates), 11 were selected according to the following criteria: (1) male or female children or adolescents at a mean age of 18 years or younger, with obesity; (2) intervention study consisting of a physical exercise program, or a nutrition program/consultation, or medication/supplements, or some combination of them; (3) controlled or not controlled randomized and nonrandomized trials published in English; (4) results including somatic stress, inflammation, or dysmetabolism indicators as Mean Values with Standard Deviation (SD). The detailed steps of the systematic article search and selection process are presented in a flow chart in Figure 2.

### 2.3. Data Extraction

The following data were extracted from each study: the first author’s name and the year of publication, the number, age, and gender of the participants, the duration of the study, the type of intervention, and the type of the trial (Table 2). The parameters gathered included body measurements (BMI, Waist Circumference, Body Fat percentage), dysmetabolism indicators (Glucose, Insulin, HOMA-IR, HDL, LDL, Triglycerides, Adiponectin, Leptin, Systolic, and Diastolic Blood Pressure), and inflammation markers (CRP, TNF-a, IL-6). For each parameter, a random-effects model on Cohen’s d distance meta-analysis was tested to compare the pooled weighted means at the endpoint of the studies to those at baseline or of control groups. Another author (Τ.A.) compared the extraction forms; all differences were reviewed, discussed, and corrected. The studies that met all inclusion criteria but did not report sufficient quantitative data were considered for a qualitative analysis only.

### 2.4. Statistical Analysis

All analyses were performed using IBM SPSS Statistics Version 28.0.1.0 (142). Each study’s effect size and 95% confidence intervals (95%CI) were calculated using inverse variance weighting. The number of patients, mean values, and standard deviation were considered for each treatment group. When quantitative data regarding outcomes were reported in different units, they were converted into the most used units. For those studies whose participants were allocated into three groups (e.g., a control group and two intervention groups with different protocols), the data of every intervention group were independently compared with the data of the control group. The random-effects model was chosen, and heterogeneity was assessed using the Q statistics. The heterogeneity variance T2 was measured. I2 was used to express the heterogeneity as a percentage. A Q value with a significance of *p* less than or equal to 0.05 (two-tailed) was considered significant to reject the null hypothesis. Forest and funnel plots were generated to illustrate the study-specific effect sizes along with the 95% CI.

Generally, when conducting a meta-analysis on heterogeneous data for obesity, several errors or challenges can arise due to the diversity of studies included. Heterogeneous data may include studies that differ significantly in terms of population characteristics, interventions, outcome measures, or study designs. Comparing such diverse studies can lead to inappropriate conclusions and misinterpretations of the overall effect. Failing to acknowledge or properly address the heterogeneity among studies can result in a misleading summary effect estimate. Also ignoring heterogeneity may lead to a false sense of precision and can undermine the validity of the meta-analysis. Using a fixed-effect model to analyze heterogeneous data can lead to inaccurate effect estimates, as this model assumes that the true effect size is the same across all included studies. Heterogeneous data can increase the risk of publication bias, where studies with significant results are more likely to be published than those with non-significant results. Failing to account for publication bias can distort the overall effect estimate. To address heterogeneity, researchers may conduct subgroup analyses. However, subgroup analyses should be interpreted cautiously, as they can lead to false conclusions if not pre-specified and justified based on prior knowledge. Another issue can arise when combining data from studies with different measurement scales or units. Inappropriate data transformations can lead to erroneous effect estimates and misinterpretations. Finally, it is important to carefully consider the study quality and potential sources of heterogeneity before its exclusion. Without exploring and understanding the potential sources of heterogeneity, researchers may miss valuable insights into the factors that influence the variability of a study outcomes [49,50,51,52,53].

To address these errors and challenges, it is essential to conduct a thorough assessment of heterogeneity, consider appropriate statistical methods (e.g., random-effects models), explore potential sources of variability, and interpret the results within the context of the study’s limitations and the diversity of the included studies.

## 3. Results

A bibliogram is a linguistics construct with the distinctive property that it is not the primary product of speaking or writing but, rather, a secondary or derivative product that emerges only through analysis [23]. Figure 3a–c shows the bibliogrammatic networks based on lexical unit co-occurrences in the collection of abstracts representing the landscape of papers resulting from each database.

According to these networks, second-generation search strings were composed and tried on each database once again (the search results and the number of papers for the second-generation search strings are available in Appendix A).

### 3.1. Study Selection and Characteristics

The literature search, after the removal of duplicates, identified 53 articles containing the indicated keywords. Twenty-two studies had their full text analyzed based on the inclusion criteria. As a result of this selection, 11 studies [38,39,40,41,42,43,44,45,46,47,48], enrolling 649 children or adolescents living with obesity or overweight, with mean ages from 8 to 16 years old, were included in the qualitative synthesis. The detailed steps of the systematic article search and selection process are presented in the flow chart in Figure 2.

Of the 11 intervention studies, 7 were randomized controlled trials, 3 were randomized but not controlled, and 1 was a cohort study, with a duration between 3 and 12 months and a publication date between 2005 and 2021. In total, 10 studies included interventions with diet and exercise, 2 of them with supplement or drug addition to the scheme, and 1 study focused on exercise alone without diet changes. The general study characteristics are summarized in Table 2. The markers of inflammation and dysmetabolism that were examined were Glucose, Insulin, HOMA-IR, HDL, LDL, Triglycerides, Adiponectin, Leptin, Systolic and Diastolic Blood Pressure, CRP, TNF-A, and IL-6. The eligible analysis indicators per study are reported in Table 3. The data reported by Farpour–Lambert et al., 2019 [43], were not suitable for meta-analysis, since not enough information was provided by the authors to calculate standard deviations, but they were kept for the qualitative analysis.

### 3.2. Meta-Analysis

#### 3.2.1. Anthropometric Parameters

Regarding the anthropometric parameters, BMI and Waist Circumference were not affected significantly by the intervention. The only parameter that was significantly reduced in the Intervention Group (IG) was Body Fat, as shown in Figure 4.

#### 3.2.2. Indicators of Dysmetabolism

The reduction in glucose levels was not statistically significant for the IG, except in the subgroup analysis focusing on randomized controlled trials, where the glucose levels were significantly decreased for the IG over the Control Group (CG) [Effect Size = −0.83, 95%-CI (−1.651, −0.004), *p* = 0.05], as shown in Figure 5. In the IG, there was a statistically significant reduction in insulin levels [Effect Size = −0.73, 95%-CI (−1.30, −0.13), *p* = 0.01] compared to the CG, as shown in Figure 6. HOMA-IR was also statistically significant in the subgroup of randomized controlled trials but not in the overall analysis [Effect Size = −1.38, 95%-CI (−1.91, −0.85), *p* < 0.001], as shown in Figure 7.

There was no significant change in lipid levels (HDL, LDL, TG) and SBP in the overall or subgroup analyses, but a statistically significant reduction was noted in DBP for the IG [Effect Size = −0.32, 95%-CI (−0.55, −0.09), *p* = 0.01], as shown in Figure 8.

#### 3.2.3. Indicators of Inflammation

In the IG, the adiponectin levels increased compared to the CG [Effect Size = 0.48, 95%-CI (0.01, 0.94), *p* = 0.04], as shown in Figure 9, and the CRP levels were statistically significantly reduced in the subgroup of randomized controlled trials but not in the overall analysis [Effect Size = −0.44, 95%-CI (−0.87, −0.001), *p* = 0.05], as shown in Figure 10. The leptin levels showed no significant change between the IG and the CG in the overall analysis. All indicators and *p* values resulting from the meta-analysis are reported in Table 4.

IL-6 levels were reported in three studies [39,40,44], with controversial results. In Balagopal et al., 2005 [39], and Kahhan et al., 2021 [40], the IL-6 levels were significantly reduced in the intervention group vs. the control group (*p* < 0.05), but in Rynders et al., 2012 [18], there was no significant change in the IL-6 levels between the intervention and the control group. The TNF-alpha levels were reported in one study [44], with no significant changes between the intervention and the control group (*p* > 0.05).

## 4. Discussion

Herein, an effort was made to shed light on the impact that diet/exercise interventions can achieve on markers related to inflammation and dysmetabolism in children living with obesity or overweight. Our findings indicate that any intervention aiming to reduce excess body fat (diet, exercise, supplements/drugs) in children and adolescents living with obesity or overweight can lead to significant changes in a variety of markers, mainly by re-establishing homeostasis even if the change is subtle.

In a similar meta-analysis, Sirico F. et al., 2018 [54], studied the effects of physical exercise on adiponectin, leptin, and inflammatory markers (IL-6, CRP, TNF-alpha) in children with obesity. The authors of the study reported significant changes in both adiponectin and leptin levels in favor of the intervention group. Also the IL-6 levels were significantly reduced. A trend towards a reduction was also observed in the CRP levels, although no effect on the TNF-alpha levels was reported, due to contradictory results. However, there are major differences between our study and this prior meta-analysis, mainly in the selection of the studies to be analyzed. Notably, our study had an open plan and did not focus only on randomized controlled trials; in addition, there is a difference in the nature of the scientific question, which, in our study, included every available intervention. Finally, herein we investigated not only inflammatory but also dysmetabolism markers.

Another meta-analysis by Schwingshackl L. et al., 2015 [55], focused on the effect of a low glycemic load diet on several risk factors in children and adolescents living with obesity or overweight. The parameters considered were body weight, body mass index, z-score of the body mass index, fat mass, fat-free mass, height, waist circumference, hip circumference, waist-to-hip ratio, total cholesterol, LDL-cholesterol, HDL- cholesterol, triglycerides, diastolic and systolic blood pressure, fasting serum glucose, fasting serum insulin, HOMA-IR index, glycosylated hemoglobin, and C-reactive protein. Significant changes were observed in the triglyceride levels and HOMA-IR index in the low-glycemic diet group. This meta-analysis was also focused on randomized controlled trials. Unlike our own, the intervention included only diet changes and did not consider some of the markers we monitored in our study.

At this point, it is crucial to focus the attention on adiponectin’s and leptin’s roles as adipokines involved in several biochemical pathways and thus inevitably modulating metabolic processes and inflammatory responses. Adiponectin’s metabolic effects include a reduction in glucose production and an increase in insulin sensitivity and energy expenditure, while leptin signals energy sufficiency and also increases energy expenditure. Regarding inflammation, adiponectin has anti-inflammatory properties, as it promotes the reduction of TNF-alpha production in macrophages, and leptin has proinflammatory properties, as high levels of leptin activate monocytes and macrophages to produce IL-6 and TNF-alpha. [56,57]. In children with obesity or overweight, the adiponectin levels are significantly lower compared to those in normal-weight children, while the leptin levels are increased [58]. The importance of our findings regarding adiponectin and leptin (a trend towards a reduction, driven by the randomized controlled trials subgroup), lies in the fact that their levels changed preferentially in the intervention group after any type of intervention, even if the BMI was not significantly altered (however, there was a significant change in body fat).

Another significant finding was that reduced insulin levels were observed after intervention, and the trend towards a reduction in both HOMA-IR and glucose levels was driven by randomized controlled trials. It is common among individuals living with obesity or overweight for insulin levels to be elevated, and this is also associated with insulin resistance. Insulin resistance (IR) is a pathological condition supporting several dysmetabolic conditions including obesity and type 2 diabetes (T2D), dyslipidemia, atherosclerosis, polycystic ovarian syndrome (PCOS), and non-alcoholic fatty liver disease (NAFLD). In children and adolescents with obesity, of any age, a strong association between IR and a higher prevalence of the components of the metabolic syndrome (MS) was observed; therefore a higher cardiovascular risk is predicted in these subjects [59].

Finally, the significant reduction in systolic blood pressure that we observed needs to be mentioned, as an elevated blood pressure is firmly reported in children and adolescents living with obesity or overweight and contributes to the occurrence and severity of hypertension along with an increase in cardiovascular risk. The duration of hypertension affects the risk of end-organ damage; so, it is of uttermost importance to restrain and control the blood pressure at a young age. Considering that the global prevalence of childhood hypertension is rising along with the prevalence of overweight and obesity [60], every measure taken against blood pressure elevation is important.

The results of the current meta-analysis should be considered with caution, as some limitations can be identified. Firstly, only a few studies focused on the several outcome parameters chosen for the data synthesis were available. This fact could potentially explain the lack of statistical significance for the differences observed in various outcome parameters between the intervention and the control group and the occurrence of tendencies towards an improvement in the groups. Secondly, all the included studies had a small sample size (about 100 or less). Another limitation concerns the duration and intensity of the interventions, as the several protocols used followed neither the same time frame for the completion of the study nor the same exercise duration. For example, more significant differences could have been observed between the control and the intervention groups with a longer study duration or exercise time. Nevertheless, it was important to include studies with different durations to increase the sample size and improve the statistical viability and the credibility of our scientific arguments. Generally, the duration of the interventions in childhood/adolescence obesity/overweight studies is crucial for understanding their short- and long-term impacts on various physiological, behavioral, and developmental aspects. It influences the ability to detect sustained effects, assess developmental interactions, study metabolic changes, observe behavioral adaptations, and determine the overall efficacy and feasibility of the interventions. Even though these limitations are common in interventional studies including exercise protocols, they affect the validity of the results.

Another concern arising from the study of leptin and adiponectin in overweight/obesity in children/adolescents regards the different expression of these adipokines according to sex and age; in fact, hormonal differences between boys and girls and between prepubertal and postpubertal individuals of both sexes are major determinants of the levels of plasma adipokines [58]. The studies included in our meta-analysis were not specifically designed nor performed any special calculation of the results considering the mentioned issues.

## 5. Conclusions

All in all, the current meta-analysis provides evidence of the beneficial effect of a lifestyle intervention (diet, exercise, supplements/drugs) in children and adolescents with obesity or overweight. With a significant reduction in Body Fat (although not in BMI or Waist Circumference), an increase in the adiponectin levels, a reduction in the circulating insulin levels and in diastolic blood pressure, and a trend towards a reduction of circulating leptin and glucose levels and HOMA-IR, our findings corroborate the hypothesis that lifestyle interventions could reduce overweight-/obesity-associated systemic inflammation and dysmetabolism. Considering the discussed limitations of this analysis, further studies are necessary to confirm our findings.

As overweight and obesity among children and adolescents have become a major pandemic, it is very important to sensitize not only individuals but also public health stakeholders. As obesity and overweight at a young age most commonly persist in adults, causing several health complications, it is extremely important to address this problem as early as possible.

## Figures and Tables

**Figure 1 jpm-13-01322-f001:**
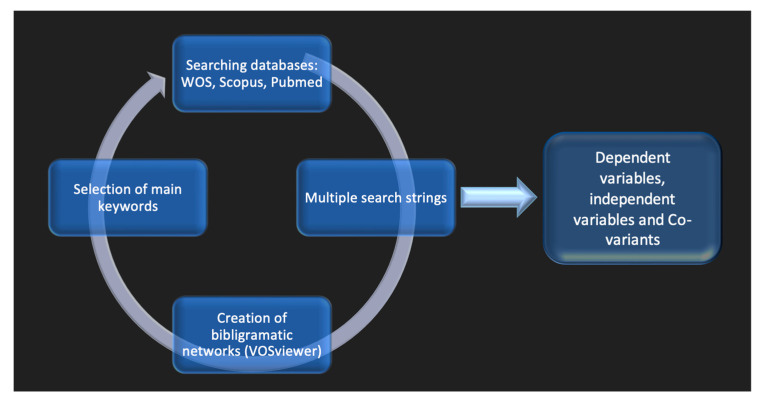
Methodology flow chart.

**Figure 2 jpm-13-01322-f002:**
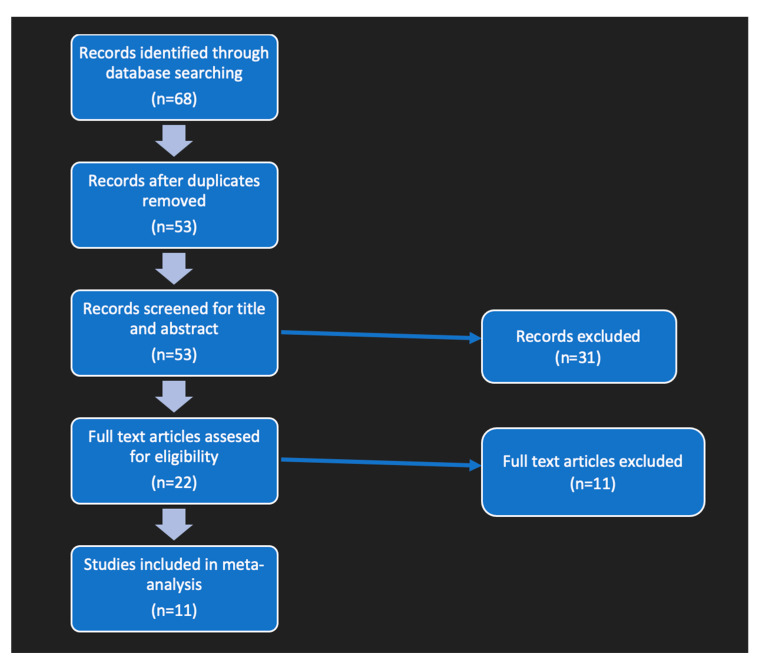
Flow diagram of the study selection process.

**Figure 3 jpm-13-01322-f003:**
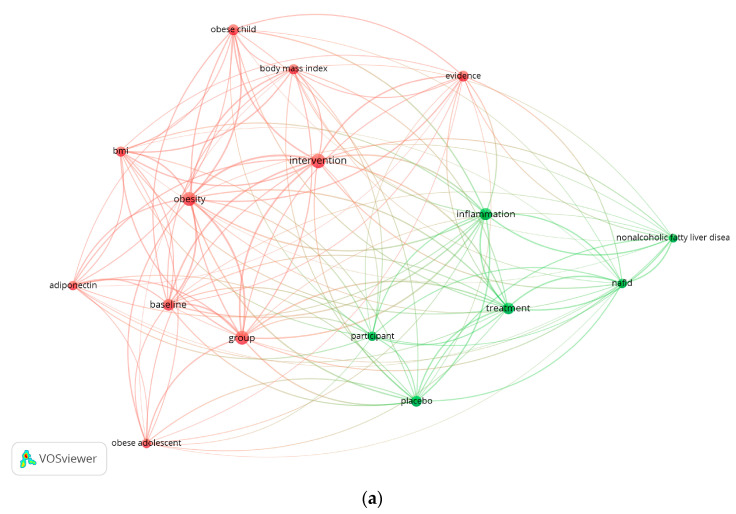
Keywords network resulting from certain search string in (**a**) Web of science, (**b**) Scopus, and (**c**) PubMed.

**Figure 4 jpm-13-01322-f004:**
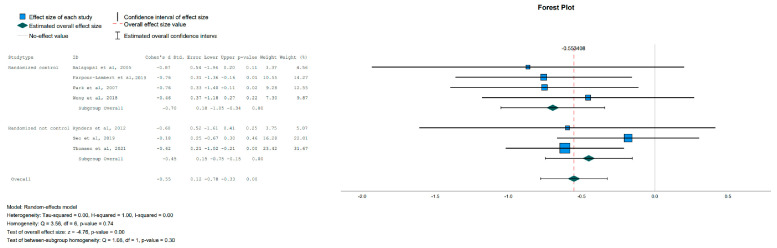
Forest plot showing the pooled standardized effect size with 95% CI for Body Fat for 7 studies divided into 2 subgroups [39,40,41,42,43,46,47]. For each study, the shaded square represents the point estimate of the intervention effect. The horizontal line joins the lower and the upper limits of the 95% CI of these effects. The shaded square area reflects the study’s relative weight in the respective meta-analysis. The diamond at the bottom of the graph represents the estimated overall effect size with the 95% CI for the seven study groups.

**Figure 5 jpm-13-01322-f005:**
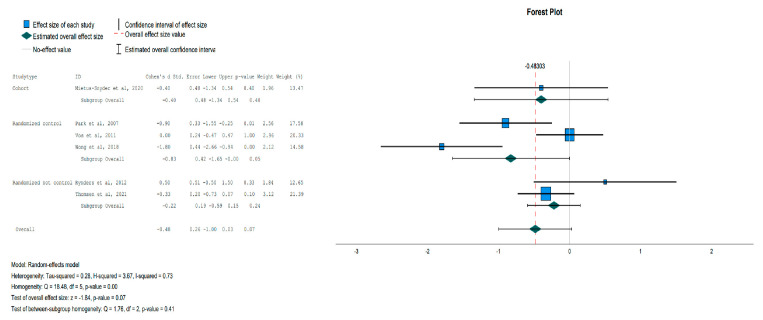
Forest plot showing the pooled standardized effect size with 95% CI for Glucose for 6 studies divided into 3 subgroups [38,40,41,42,45,46]. For each study, the shaded square represents the point estimate of the intervention effect. The horizontal line joins the lower and the upper limits of the 95% CI of these effects. The shaded square area reflects the study’s relative weight in the respective meta-analysis. The diamond at the bottom of the graph represents the estimated overall effect size with the 95% CI for the seven study groups.

**Figure 6 jpm-13-01322-f006:**
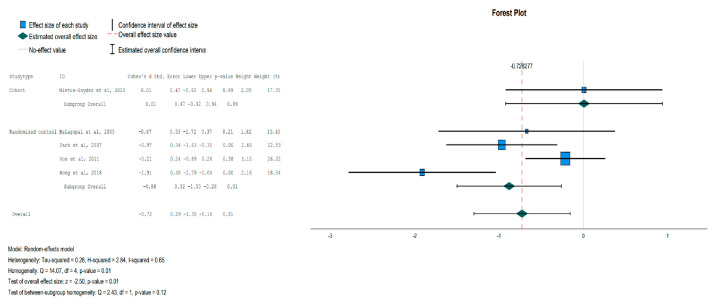
Forest plot showing the pooled standardized effect size with 95% CI for Insulin for 5 studies divided into 2 subgroups [38,39,42,45,46]. For each study, the shaded square represents the point estimate of the intervention effect. The horizontal line joins the lower and the upper limits of the 95% CI of these effects. The shaded square area reflects the study’s relative weight in the respective meta-analysis. The diamond at the bottom of the graph represents the estimated overall effect size with the 95% CI for the seven study groups.

**Figure 7 jpm-13-01322-f007:**
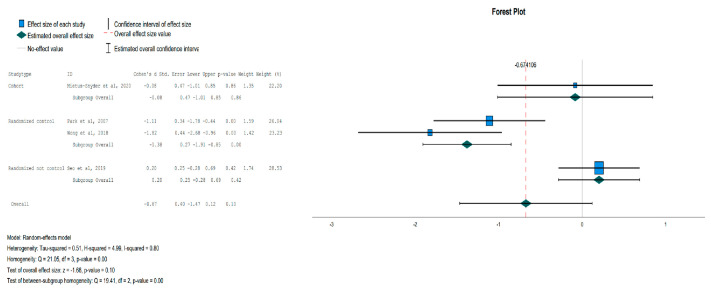
Forest plot showing the pooled standardized effect size with 95% CI for HOMA-IR for 4 studies divided into 3 subgroups [42,45,46,47]. For each study, the shaded square represents the point estimate of the intervention effect. The horizontal line joins the lower and the upper limits of the 95% CI of these effects. The shaded square area reflects the study’s relative weight in the respective meta-analysis. The diamond at the bottom of the graph represents the estimated overall effect size with the 95% CI for the seven study groups.

**Figure 8 jpm-13-01322-f008:**
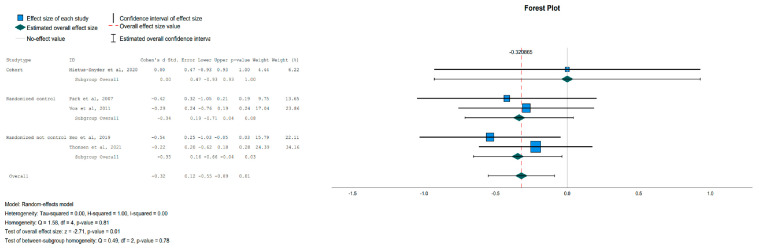
Forest plot showing the pooled standardized effect size with 95% CI for DBP for 5 studies divided into 3 subgroups [38,41,45,46,47]. For each study, the shaded square represents the point estimate of the intervention effect. The horizontal line joins the lower and the upper limits of the 95% CI of these effects. The shaded square area reflects the study’s relative weight in the respective meta-analysis. The diamond at the bottom of the graph represents the estimated overall effect size with the 95% CI for the seven study groups. DBP: Diastolic Blood Pressure.

**Figure 9 jpm-13-01322-f009:**
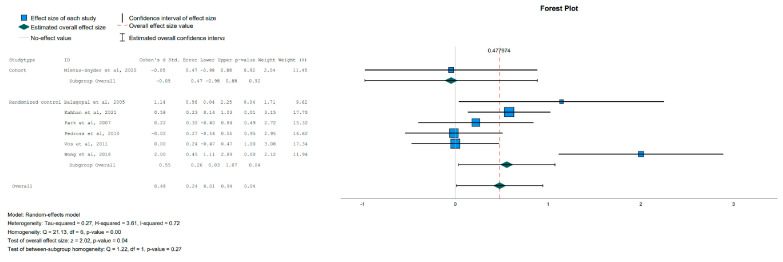
Forest plot showing the pooled standardized effect size with 95% CI for Adiponectin for 7 studies divided into 2 subgroups [38,39,42,44,45,46,48]. For each study, the shaded square represents the point estimate of the intervention effect. The horizontal line joins the lower and the upper limits of the 95% CI of these effects. The shaded square area reflects the study’s relative weight in the respective meta-analysis. The diamond at the bottom of the graph represents the estimated overall effect size with the 95% CI for the seven study groups.

**Figure 10 jpm-13-01322-f010:**
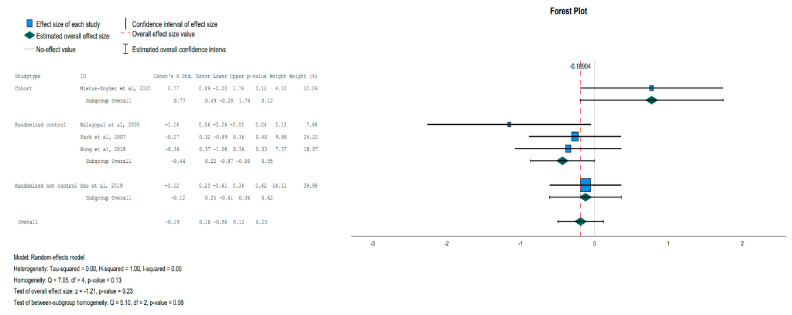
Forest plot showing the pooled standardized effect size with 95% CI for CRP for 5 studies divided into 3 subgroups [39,42,45,46,47]. For each study, the shaded square represents the point estimate of the intervention effect. The horizontal line joins the lower and the upper limits of the 95% CI of these effects. The shaded square area reflects the study’s relative weight in the respective meta-analysis. The diamond at the bottom of the graph represents the estimated overall effect size with the 95% CI for the seven study groups. CRP: C-Reactive Protein.

**Table 1 jpm-13-01322-t001:** Abbreviations: HOMA-IR, Homeostatic Model Assessment for Insulin Resistance; HDL, High-Density Lipoprotein; LDL, Low-Density Lipoprotein; TG, Triglycerides; SBP, Systolic Blood Pressure; DBP, Diastolic Blood Pressure; CRP, C-Reactive Protein; BMI, Body Mass Index; WC, Waist Circumference.

Dependent Variable	Independent Variable	Co-Variants
Leptin, Adiponectin, Glucose, Insulin, HOMA-IR, HDL, LDL, TG, SBP, DBP, CRP	BMI, WC, Body Fat	Age, Sex, Duration, Intervention protocol, Trial protocol

**Table 2 jpm-13-01322-t002:** General characteristics of the studies, Refs. [38,39,40,41,42,43,44,45,46,47,48], included in the meta-analysis.

	Population (n)	Age	Gender (M/F)	Duration (Months)	Intervention	Type of Trial
Vos et al., 2011	113	13.2	51/62	12	Diet and Exercise and Drugs/Supplements	Randomized controlled
Balagopal et al., 2005	21	15.8	11/10	3	Diet and Exercise	Randomized controlled
Rynders et al., 2012	16	14.2	7/9	6	Diet and Exercise and Drugs/Supplements	Randomized not controlled
Thomsen et al., 2021	99	12.0	45/54	12	Diet and Exercise	Randomized not controlled
Wong et al., 2018	30	15.3	0/30	3	Exercise	Randomized controlled
Farpour-Lambert et al., 2019	74	9.6	38/36	12	Diet and Exercise	Randomized controlled
Kahhan et al., 2021	87	10	29/58	12	Diet and Exercise	Randomized controlled
Mietus-Snyder et al., 2020	18	15.5	Ν/A	Ν/A	Diet and Exercise and Drugs/Supplements	Cohort
Park et al., 2007	40	14.2	0/40	3	Diet and Exercise	Randomized controlled
Seo et al., 2019	70	12.5	45/25	4	Diet and Exercise	Randomized not controlled
Pedrosa et al., 2010	61	8.7	27/34	12	Diet and Exercise	Randomized controlled

**Table 3 jpm-13-01322-t003:** Indicators of inflammation and dysmetabolism included in the selected studies [38,39,40,41,42,43,44,45,46,47,48] that were eligible for analysis.

	Glucose	Insulin	HOMA-IR	HDL	LDL	TG	SBP	DBP	CRP	Leptin	Adiponectin	TNF-A	IL-6
Balagopal et al., 2005		✓							✓		✓		✓
Farpour-Lambert et al., 2019													
Kahhan et al., 2021									✓	✓	✓	✓	✓
Mietus-Snyder et al., 2020	✓	✓	✓	✓	✓	✓	✓	✓	✓		✓		
Park et al., 2007	✓	✓	✓	✓	✓	✓	✓	✓	✓	✓	✓		
Pedrosa et al., 2010		✓	✓	✓		✓	✓	✓	✓	✓	✓		
Rynders et al., 2012		✓											✓
Seo et al., 2019			✓	✓	✓	✓	✓	✓	✓				
Thomsen et al., 2021	✓						✓	✓		✓			
Vos et al., 2011	✓	✓		✓		✓	✓	✓	✓		✓		
Wong et al., 2018	✓	✓	✓						✓	✓	✓		

**Table 4 jpm-13-01322-t004:** Pooled estimates of the effect size for the results of the Intervention Group compared to the Control Group, * statistically significant.

Outcome Parameter	Standardized Effect Size	95% CI	*p* Value	No of Studies	Sample Size	I2 (%)
BMI (kg/m^2^)	−0.374	(−0.836, −0.089)	0.113	6	250	64
Body Fat (%)	−0.553	(−0.781, −0.325)	<0.001	7	317	0
WC (cm)	−0.349	(−0.869, 0.170)	0.187	5	205	67
Glucose (mmol/L)	−0.483	(−0.097, 0.031)	0.65	6	272	73
Insulin (mU/L)	−0.728	(−1.299, 0.157)	0.01 *	5	172	65
HOMA-IR	−0.674	(−1.468, 0.119)	0.1	4	158	80
HDL (mmol/L)	−0.019	(−0.303, 0.265)	0.9	4	197	0
LDL (mmol/L)	−0.207	(−0.563, 0.150)	0.25	3	128	0
Triglycerides (mmol/L)	0.712	(−0.889, 2.314)	0.36	4	197	96
SBP (mmHg)	−0.225	(−0.475, 0.024)	0.08	4	256	0
DBP (mmHg)	−0.321	(−0.553, −0.089)	0.01 *	5	296	0
CRP (mg/L)	−0.189	(−0.496, 0.118)	0.2	5	173	0
Leptin (ng/mL)	−0.457	(−1.135, 0.222)	0.19	4	250	84
Adiponectin (mg/L)	0.478	(0.013, 0.943)	0.04 *	7	318	72

## Data Availability

The data used to support the findings of this study are available upon reasonable request. Requests for access to the data can be directed to Konstantina Dragoumani at konstantinadragoumani@gmail.com. Due to privacy and ethical considerations, some restrictions may apply to the availability of certain sensitive or confidential data.

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
