# Peer review of "Childhood and Adolescent Obesity with Somatic Indicators of Stress, Inflammation, and Dysmetabolism before and after Intervention: A Meta-Analysis"

_jpm, 2023, doi:10.3390/jpm13091322_

Round 1

Reviewer 1 Report

It is recommended not to use the word obese. Please correct the entire article for obesity, or people living with obesity.

It is convenient for the discussion to describe the types of diets and the type of exercise that the study population received in the interventions, since it may be an important point to consider. The time of the interventions in the studies are important limitations to also consider in the discussion, possibly due to the impact that can be found in the modification of the phenotypes studied.

Author Response

It is recommended not to use the word obese. Please correct the entire article for obesity, or people living with obesity.

  • Thank you very much for your recommendation. We also favor not to use the word obese. In this direction we have replaced all “obese” words with “obesity” or “people living with obesity”.

It is convenient for the discussion to describe the types of diets and the type of exercise that the study population received in the interventions, since it may be an important point to consider.

  • This is a valid point, and we thank the Reviewer for it. As a matter of fact, initially we thought about it and we drafted a supplementary table with all the different parameters that were used in each study. However, we decided not to include this table because of the heterogeneity of the different parameters in each study, which would be confusing to the reader. We have paid a lot of attention to properly cite all pertinent papers, add relevant references and ensure that the interested reader will be properly directed to the right source to retrieve the types of diets and the type of exercise that the study population received with each intervention.

The time of the interventions in the studies are important limitations to also consider in the discussion, possibly due to the impact that can be found in the modification of the phenotypes studied.

  • Of course, duration of each intervention may be important and this is the reason why we have included a table in the methods of the paper mentioning the range and average time length of the interventions. We will give extra credit in time limitations at the discussion.

Reviewer 2 Report

The authors have provided a study that assesses stress, Inflammation, and metabolic dysfunction before and after intervention in children and young adolescents with obesity. They have conducted a meta-analysis of the results obtained from their search strategies. However, there are some issues with the text that are evident within the first two sections of the article.

The introduction does not provide adequate background on the topic. There are general comments provided but there is no clear evidence provided or described as to why the topics needs to be explored. It would be beneficial to outline and provide worked/studied examples of the complications associated with obesity and the potential issues that can result from chronic low-grade systemic inflammation in general and then outline the current issue/s in this setting. The section on the methodology reasoning is the most specific component of this section and is somewhat beneficial to the reader. However, it just o be refined more with the negatives and challenges associated with current methods specifically elaborated on further. Please ensure that examples are provided that are referenced regarding increased errors, issues, etc. rather than generalising. Cochrane have an entire section on this and is why protocols for this have to be published prior to search strategies are implemented to avoid these issues. Also, do the PRISMA guidelines not state why the protocol needs to be followed when searching systematically in order to conduct a meta-analysis to avoid conflicts, errors, bias, etc.? Further, there are a distinct lack of references used in this document, particularly in the introduction. There are very matter of fact statements and statistics that are provided but there are no references provided. Please update this to ensure all statements are referenced accordingly. There is also a common theme of non-academic language used throughout the text that could be improved on to ensure words used are done so in the correct context (e..g line 51 - devastating, line 53 - hard outcome, 70 - dear). Finally, the aim of this article is not made clear. 

It would also be prudent to outline what ‘interventions’ are being explored in the introduction. The title states lifestyle interventions, but there is no discussion or mention of this in the introduction. Please update to include 1 – 2 paragraphs outlining possible interventions and what is meant by lifestyle interventions and whether or not obesity is impacted by poor lifestyle choices, etc. (yes it is, but there is no evidence provided to warrant this in the introduction and it is a key criteria for the paper). 

The methods section is not clear. Were the databases searched for systematically? It seems unlikely that only 68 studies/findings would be obtained from the key words search terms if a systematic strategy was applied. It is not clear how the interventions were searched for and subsequently selected. There are no definitions provided about lifestyle interventions and what constitutes these. For example physical exercise has many sub-categories and it would be prudent to outline what these were and how they were included as there are many studies that have looked at interventions in obese children and adolescents that have taken some blood work specific to the search criteria. The search method does not appear to be systematic or organised sufficiently and it is not specifically clear what the exclusion criteria were, thus making it difficult to draw adequate conclusions from the data generated. 

Please see comment above in the comments and suggestions above regarding English use. This could be improved by simple editing and re-reading to ensure the use of non-academic terms are removed. 

Author Response

The authors have provided a study that assesses stress, Inflammation, and metabolic dysfunction before and after intervention in children and young adolescents with obesity. They have conducted a meta-analysis of the results obtained from their search strategies. However, there are some issues with the text that are evident within the first two sections of the article.

The introduction does not provide adequate background on the topic. There are general comments provided but there is no clear evidence provided or described as to why the topics needs to be explored. It would be beneficial to outline and provide worked/studied examples of the complications associated with obesity and the potential issues that can result from chronic low-grade systemic inflammation in general and then outline the current issue/s in this setting.

  • Thank you very much for your valuable comments and constructive criticism. It goes without saying that especially nowadays studying child obesity is extremely important. However, and in relation to the essence and the take home message of our article we see the point that you are trying to make and we agree. In this direction we have revised and partially restructured the Introduction. 

The section on the methodology reasoning is the most specific component of this section and is somewhat beneficial to the reader. However, it just o be refined more with the negatives and challenges associated with current methods specifically elaborated on further. Please ensure that examples are provided that are referenced regarding increased errors, issues, etc. rather than generalising.

  • So this section had been revised to read better and examples have been added.

Further, there are a distinct lack of references used in this document, particularly in the introduction. There are very matter of fact statements and statistics that are provided but there are no references provided. Please update this to ensure all statements are referenced accordingly.

  • This is a comment raised by the other Reviewer as well. We have added quite a few references to strengthen all major points and statements included in our article.

There is also a common theme of non-academic language used throughout the text that could be improved on to ensure words used are done so in the correct context (e.g., line 51 - devastating, line 53 - hard outcome, 70 - dear).

  • Thank you for spotting that. We confirm we have revised and updated the manuscript accordingly.

Finally, the aim of this article is not made clear. It would also be prudent to outline what ‘interventions’ are being explored in the introduction. The title states lifestyle interventions, but there is no discussion or mention of this in the introduction. Please update to include 1 – 2 paragraphs outlining possible interventions and what is meant by lifestyle interventions and whether or not obesity is impacted by poor lifestyle choices, etc. (yes it is, but there is no evidence provided to warrant this in the introduction and it is a key criteria for the paper).

  • The other Reviewer somehow raised the same point. We have revised, therefore, the Introduction and we have ensured that the meaning of lifestyle interventions is properly stated and referenced.

The methods section is not clear. Were the databases searched for systematically? It seems unlikely that only 68 studies/findings would be obtained from the key words search terms if a systematic strategy was applied. It is not clear how the interventions were searched for and subsequently selected. There are no definitions provided about lifestyle interventions and what constitutes these. For example, physical exercise has many sub-categories and it would be prudent to outline what these were and how they were included as there are many studies that have looked at interventions in obese children and adolescents that have taken some blood work specific to the search criteria. The search method does not appear to be systematic or organised sufficiently and it is not specifically clear what the exclusion criteria were, thus making it difficult to draw adequate conclusions from the data generated.

  • The methods section has been revised and it is clear now that this is a systematic review. Also, quite a few references have been added and it is now clear what lifestyle intervention is (as this is a comment raised before). The take home message has also been made clearer. We would like to thank the Reviewer for taking the time and effort to provide us with constructive criticism that overall improved our article and undoubtedly promoted science.

Many thanks for resubmitting your revised paper, and thanks for the effort made to improve the paper and address the issues raised by the reviewers’ comments and feedback.

  • Thank you.

Round 2

Reviewer 2 Report

Thank you for addressing the comments listed previously. However, the primary concern here is that the methods section has not been updated (granted the introduction and results section have been), but more results were generated without significantly changing the methods and search requirements. Could the authors please comment and ensure that all the studies listed are correct and are similar to that of a robust search strategy such as the PRISMA guidelines for conducting systematic studies? 

The English is much improved, but minor editing would be beneficial in this instance to remove all non-academic language. 

Author Response

Thank you for addressing the comments listed previously.

  • Thanks so much for the previous round of suggestions and criticism. Helped to improve the article significantly.

However, the primary concern here is that the methods section has not been updated (granted the introduction and results section have been), but more results were generated without significantly changing the methods and search requirements. Could the authors please comment and ensure that all the studies listed are correct and are similar to that of a robust search strategy such as the PRISMA guidelines for conducting systematic studies?

  • Indeed, a lot of effort was paid to revise the intro and results sections (as per reviewer’s request). Having said that, we must stress that although the tight revision time allowed by the editor, we did add two extra paragraphs in the methods section too. However, in this instance the reviewer is requesting comments in methodology ensuring that the search strategy that we used herein to conduct the systematic study is robust (as in PRISMA). In this direction, we have added an extra paragraph describing the steps followed and decisions made and adopted following state of the art standard of procedures to ensure viability, reliability and robustness in regards to our search strategy.

The English is much improved, but minor editing would be beneficial in this instance to remove all non-academic language. 

  • Thanks. Few more edits were made, and the manuscript was also reviewed by a close collaborator of ours who is a native English speaker.